# Laboratory-Controlled Experiments Reveal Microbial Community Shifts during Sediment Resuspension Events

**DOI:** 10.3390/genes13081416

**Published:** 2022-08-09

**Authors:** Alexis DesRosiers, Nathalie Gassama, Cécile Grosbois, Cassandre Sara Lazar

**Affiliations:** 1Department of Biological Sciences, University of Québec at Montréal, UQAM, 141 Avenue du Président-Kennedy, Montreal, QC H2X 1Y4, Canada; 2GéHCO Géo-Hydrosystèmes Continentaux, Université de Tours, 37200 Tours, France

**Keywords:** dam freshwater, dam sediment, sediment leaching, archaea, bacteria

## Abstract

In freshwater ecosystems, dynamic hydraulic events (floods or dam maintenance) lead to sediment resuspension and mixing with waters of different composition. Microbial communities living in the sediments play a major role in these leaching events, contributing to organic matter degradation and the release of trace elements. However, the dynamics of community diversity are seldom studied in the context of ecological studies. Therefore, we carried out laboratory-induced leaching experiments, using sediments from the Villerest dam reservoir (Villerest, France). To assess whole microbial community diversity, we sequenced the archaeal and bacterial 16S rRNA genes using Illumina MiSeq. Our results suggest that the degree of dissolved oxygen found in the water during these resuspension episodes influenced community dynamics, with anoxic waters leading to drastic shifts in sedimentary communities compared to oxic waters. Furthermore, the release of microbial cells from sediments to the water column were more favorable to water colonization when events were caused by oxic waters. Most of the bacteria found in the sediments were chemoorganotrophs and most of the archaea were methanogens. Methylotrophic, as well as archaeal, and bacterial chemoorganotrophs were detected in the leachate samples. These results also show that organic matter degradation occurred, likely participating in carbonate dissolution and the release of trace elements during freshwater resuspension events.

## 1. Introduction

In subsurface environments such as soils and sediments, microorganisms play a significant role in the weathering of minerals [1,2]. Indeed, microbial activity can enhance mineral dissolution by releasing protons and organic compounds [3,4,5], or by redox reactions [6,7]. For some microbial species, this liberates limiting nutrients/substrates required for their metabolism, such as P [8,9], Fe [10,11], or S [12,13]. However, microbial activity can inhibit element release by the development of a passivation layer at the mineral surface, such as an amorphous silicate layer [14], or by the adsorption of polysaccharides [15].

Sediments are formed of various solid phases, silicates, carbonates, iron, manganese or aluminum oxyhydroxides, sulfide, and particulate organic matter (POM). The latter originate from bedrock erosion and alteration, soil weathering, and in situ processes. Freshwater sediments (e.g., lakes, dams, streams) typically harbour intense microbial diversity and activity, which are correlated to the high amounts of organic matter issued from the water column’s biological activity [16,17]. In calm hydraulic conditions, water–sediment exchanges through porewater diffusion to the water column are weak but present [18,19]. During dynamic hydrological events (e.g., floods or maintenance of dam reservoirs), sediment resuspension and input of water with a different composition can change reaction conditions, inducing mineral phase dissolution/precipitation and organic matter degradation [20].

In dam reservoirs, large amounts of water are mixed during hydrodynamic events like river floods, or maintenance operations such as dredging, flushing, and sluicing. During flushing, withdrawal of anoxic hypolimnion water via a bottom outlet allows oxygenated epilimnion waters to descend modifying redox conditions at the sediment–water interface [21,22]. Depending on organic matter and suspended particle loads, floods may input oxic or anoxic waters into reservoirs [23,24]. In general, floods, flushing, and sluicing are short-term events (from 12 to 48 h). They may discharge large amounts of suspended sediment downstream [25,26], which can have adverse effects on the water quality and aquatic organisms [27].

Many leaching experiments (batch experiments) on sediments have focused on the influence of the sedimentary matrix (mineral composition, granulometry, organic matter content), pH, and redox conditions. Other studies have shown that microbial activity influences leaching by decreasing the pH [28,29], since microbes release protons and different organic compounds during organic matter degradation. However, most of these studies have focused on synthetic sediments and minerals with a limited number of microbial species [30,31].

Therefore, in this study we conducted a series of leaching experiments using real dam reservoir sediments to assess the changes of the in situ microbial community composition during resuspension events, as this has not been done before. Furthermore, both archaeal and bacterial community diversity and composition were sequenced and assessed to cover the total prokaryotic community. Studied sediments were collected from the Villerest dam reservoir (France) and the leaching experiments were carried out in the lab under oxic and anoxic conditions to simulate different environmental situations. Indeed, dam maintenance will lead to the mixing of either oxic or anoxic water with the lake water. We postulate that: (1) sediment leaching will affect sediment prokaryotic community composition, (2) sediment leaching will result in the discharge of cells into the water column, thus influencing the diversity of the water community, and (3) the presence/absence of oxygen will influence prokaryotic community composition during leaching events.

## 2. Materials and Methods

### 2.1. Study Site, Sampling, and Incubation Setup

Sediments used for this study were sampled in the Villerest dam reservoir, situated in the Upper Loire Basin (France) (Figure 1). This dam was built across the Loire River in the late 1970s and has been in operation since 1984 in order to regulate flood events, sustain low flows, and produce electricity. Regular sediment flushing operations are performed at least once every 2 years. The reservoir and surrounding area lie on crystalline rocks, sedimentary formations, and recent alluvial deposits. Several towns and associated industrial zones, plus an old coal-mining district, are located upstream from the reservoir. In addition to an elevated natural geochemical background [32], a high metal trace element enrichment of the reservoir sediments has been detected [33]. In addition, this reservoir has presented a high nutrient load since it was launched, leading to its eutrophication. Seven 50 cm-long sediment cores were collected in the deeper part of the reservoir in November 2016 with a gravity corer (UWITEC, fitted with 90-mm diameter plastic liners). The first 15 cm of each core was mixed and used for leaching experiments. In November 2017, bottom water was sampled at 30 m depth using a manual water sampler, to assess the in situ microbial diversity composition and to compare it with the community formed in the water column during the leaching experiments. Although communities may change over the course of 1 year, sampling was done during the same fall season, when community composition is typically stable in lake waters [34].

Batch experiments were conducted with open glass beakers (oxic conditions) or PP vials (anoxic conditions) over 30 days until the electrical conductivity of the leachates reached a pseudo-plateau. To accelerate the weathering of the solid phases, deionized water (DI) was used as leaching solution. No enrichment medium was added during the experiments. Sediment samples were mixed with DI at a fixed solid/solution ratio of 1/10 (12 g of sediment per 120 mL of water). Solutions were continuously agitated to stimulate resuspension events. Water and sediment aliquots were taken after 24, 120, and 720 h incubation for both the oxic and anoxic incubations. For the aerobic experiment, a 250 mL vial was used for each step and the blank. Solutions were agitated with a magnetic stirrer (100 rpm). For the anaerobic experiment, triplicates of 50 mL PP vials closed with plastic caps and equipped with inlet and outlet connectors were used. Prior to mixing with sediment, the DI water was degassed for 4.5 h with pure N_2_ gas. After filling the vials, nitrogen was added at the same time to all the subsamples for 10 min to strip dissolved O_2_ and CO_2_ from the solution. Vials were agitated on a roller shaker (40 rpm). All treatments and manipulations of subsamples were performed in a portable glove box under a N_2_ atmosphere.

### 2.2. Chemical Analysis of Water Leachate

Major element and phosphorus concentrations were determined using ICP-OES at the SARM laboratory (SARM-CRPG UMR 7358, Nancy, France). The accuracy and precision of determinations were checked at regular intervals with the international standard SLRS-5 and internal references. Typical measurement uncertainties were ±2–5% for Ca. Nitrate, and sulphate concentrations were measured using ionic chromatography with chemical suppression. Typical measurement uncertainties were ±1 × 10^−6^ mol L^−1^ for NO_3_^−^ and ± 2 × 10^−5^ mol L^−1^ for SO_4_^2−^.

To estimate the dissolved organic matter (DOM) content, DOC (dissolved organic carbon) was determined on a Shimadzu TOC/TN analyser using a non-purgeable organic carbon (NPOC) procedure that involved acidification, sparging (inorganic carbon removal), and combustion of the sample at 680 °C. Potassium hydrogenophtalate was used to prepare the standard solution. The accuracy of DOC concentration measurements was ±0.5 mg L^−1^ estimated on blank samples and a range of standard concentrations. The total dissolved nitrogen (TDN) was measured in the filtered solution using the total nitrogen unit of the Shimadzu analyser with an accuracy estimated at ±0.5 mg L^−1^. The total dissolved organic nitrogen (DON) content (ammonium + dissolved organic matter containing nitrogen = N(-III) forms) was estimated as the difference between total dissolved nitrogen (TDN) and nitrogen-nitrate (N-NO_3_) concentrations

### 2.3. DNA Extraction, Sequencing and Sequence Analysis

Six grams of the sediment samples were used to extract DNA, using the DNeasy PowerMax Soil Kit (QIAGEN, Hilden, Germany). Samples were taken prior to the incubations (0 h, 0H), after 12 h (12H), 240 h (240H), and 720 h (720H). Three grams were used for separate extractions for each sediment sample in order to produce duplicates. Ten mL of the liquid phase/leachate (overlying DI water collected at the same different timepoints as the sediment samples) was filtered using a polyethersulfone 0.2 µM filter membrane (Sartorius, Göttingen, Germany). DNA was extracted from the filters using the DNeasy PowerWater kit (QIAGEN, Germany). Given the low amount of water, no duplicate samples were extracted. The archaeal and bacterial 16S rRNA genes were sequenced using an Illumina MiSeq at the CERMO (Centre d’excellence en recherche sur les maladies orphelines, Biological Sciences department, UQAM), with the 340F (5′-CCCTACGGGGYGCASCAG-3′, [35])-915R (5′-GTGCTCCCCCGCCAATTCC-3′, [36]) and the 341F (5′-CCTACGGGAGGCAGCA-3′, [37])-785R (5′-GACTACHVGGGTATCTAATCC-3′, [38]) primer pairs, respectively. Reads were merged, processed, and analyzed using mothur v.1.47 [39], resulting in 575 bp reads for the archaea and 444 bp reads for the bacteria. Classification of 16S rRNA genes was obtained using the SILVA reference database v.138 [40]. The archaeal 16S rRNA gene datasets were further analyzed using a personal database based on the taxonomy of the Woesearchaeota [41] and the Bathyarchaeota [42]. The obtained sequences were deposited in the National Center for Biotechnology Information (NCBI) under the BioProject ID: PRJNA805122.

### 2.4. Statistical Analyses

Amplicon sequence variants (ASVs) were created using mothur. ASV tables were rarefied to 3895 reads for the archaeal, and 10,636 reads for the bacterial 16S rRNA genes, to standardize sequence efforts across samples. The Shannon diversity indices (α-diversity) were calculated using mothur. Indices were compared using a Mann–Whitney–Wilcoxon test, with the *wilcox.test* function in R v.4.1.2 [43] when comparing 2 groups of samples, and using a Kruskal–Wallis test, with the *dunnTest* function of the *FSA* package (https://cran.r-project.org/web/packages/FSA/FSA.pdf accessed on 1 August 2022), when comparing 3 groups of samples. Samples were clustered using the *pvclust* package in R [44], using a correlation distance and the average agglomeration method with 1000 bootstrap replications, on a Hellinger transformed ASV table. To test whether bacterial community composition varied significantly depending on environmental parameters, we ran permutational multivariate analyses (PERMANOVA) on the rarefied ASV tables in R, using the *adonis* function of the *vegan* package (https://cran.r-project.org/web/packages/vegan/vegan.pdf accessed on 1 August 2022). Significantly different ASVs between sample groups were identified using linear discriminant effect size analyses (lefse) using the online tool from the Huttenhower lab (https://huttenhower.sph.harvard.edu/lefse/ accessed on 1 August 2022). The FEAST method (fast expectation-maximization for microbial source tracking; [45]) was used to estimate the contribution of sediment communities to water communities, and the contribution of communities in sediment and water over the different incubation time points (i.e., the source of communities in both the sediments and water samples), using raw ASV tables without transformation.

## 3. Results

### 3.1. Geochemical Analyses in the Water Leachate over Time

Under oxic conditions, pH decreased rapidly (Figure 2a) during the first 24 h (from 6.9 to 5.1), stayed stable up to 20 days, and then decreased again to 4.3 by the end of the experiment. The main weathered mineral was a calcium carbonate according to the Ca^2+^ release (Figure 2b), which began after 24 h and continued to increase regularly from 11.93 to 89.77 mg L^−1^. Under anoxic conditions, pH slightly increased during the first 10 days (from 5.4 to 5.6) then stayed stable at around 6.4 until the end of experiment. The Ca^2+^ release followed the same trend, increasing the first 10 days (from 11.54 to 37.10 mg L^−1^) then staying stable at around 30 mg L^−1^ until day 30. For both conditions, mineral alteration, evidenced by Ca^2+^ concentration increase, seemed to start 6 to 12 h after the beginning. Under oxic conditions, pH and Ca^2+^ release varied in opposite directions, while they both increased to a pseudo-plateau under anoxic conditions.

Under oxic conditions, the initial DOC measured corresponded to that contained in porewater and it began to be consumed after 6 h (Figure 3a), increased at 10 days, and then decreased continuously. At the same time, TDN still increased, firstly composed mainly of DON then of N-NO_3_. After 6 h, the P-PO_4_ decreased until reaching a concentration lower than the detection limit at 10 days (Figure 3b). A high input of S-SO_4_ was recorded continuously until reaching a plateau after 10 days when P-PO_4_ was no longer available. Under anoxic conditions, DOC and DON concentrations were higher than under oxic conditions (Figure 3c), but N-NO_3_ was very low. The input of S-SO_4_ was half of that under oxic conditions and decreased sharply at 30 days. From the beginning of the experiments, the P-PO_4_ concentration was higher than under oxic conditions (Figure 3b). It fluctuated greatly over time. Time variations of DOC, DON, N-NO_3_, P-PO_4_, and S-SO_4_ released during these experiments can be mainly linked to organic matter degradation.

### 3.2. Archaeal 16S rRNA Gene Diversity in the Sediments and Leachate

In the sediments, for both the oxic and anoxic incubations at 3 time points (12, 240 and 720 h), there was a complete shift of the archaeal community diversity compared to the community initially present. Indeed, the S0 sample (used as a starting point for each incubation), was dominated by unclassified Archaea, Bathyarchaeota subgroup 6 (Bathy_6), and Woesarchaeota (Appendix A). We were unable to recover enough sequences for the second duplicate of the 12 h anoxic sample. In both the sediment oxic and anoxic incubations, there was an increase of the relative abundance of methanogens which dominated the samples: *Methanosarcina*, *Methanosaeta*, *Methanoregula*, *Methanocella* or Rice Cluster I (Figure 4). Bathy_6 archaea were also detected in both experiments, with a higher relative abundance in the oxic incubations.

We recovered an extremely low amount of reads in the leachate samples. Therefore, the water samples were not used for further statistical analyses. However, it is worth noting that, among these reads, Bathy_6 archaea were the highest group in the oxic samples (Appendix A). Also, we detected methanogens, candidatus Methanoperendens, cand. Nitrosotalea, and Nitrososphaeraceae, all three of which were not dominant in the sediment samples. The water sample collected from the dam lake (Dam) contained few reads as well, with *Nitrosoarchaeum* representing most of those reads.

### 3.3. Bacterial 16S rRNA Gene Diversity in the Sediments and Leachate

As for the archaeal 16S rRNA gene diversity, for both the oxic and anoxic incubations at all time points, there was a complete shift of the bacterial community diversity compared to the community initially present. The S0 sample was dominated by *Rhodanobacter*, unclassified Acidimicriia, and unc. Micropepsaceae (Appendix A). In the oxic incubations, the sediments were dominated by *Bacillus* after 12 h, then by *Sphingomonas* at 240 h, and 720 h (Figure 5). The leachate samples showed different dominant taxa, with *Sphingomonas* dominating at 12 h, *Pseudoarthrobacter* at 240 h, and *Rhodanobacter* at 720 h. In the anoxic incubations, the sediments were dominated by *Bacillus* after 12 h, then *Clostridium* at 240 h, and *Bacillus* at 720 h (Figure 5). As for the oxic incubations, the leachate samples showed different taxa, with *Rhodanobacter* dominating at 12 h, *Pseudoarthrobacter* at 240 h, and *Flavobacterium* at 720 h. The water sample collected from the dam lake (Dam) was completely different, dominated by cand. Methylopumilus, unc. Illumatobacteraceae, Xanthobacteraceae, and Gaillelales.

### 3.4. Archaeal and Bacterial α-Diversity Indices Comparison

Archaeal (Figure 6a) and bacterial sediment sample diversity indices (Figure 6b) were not significantly different between the oxic and anoxic incubation (Mann–Whitney–Wilcoxon, *p* = 0.84 for Archaea and 0.1 for Bacteria). Similarly, leachate bacterial diversity indices were not significantly different between the oxic and anoxic incubations (*p* = 0.1). However, bacterial diversity indices were significantly different between sediments and leachate sample when comparing diversity indices in oxic conditions (*p* = 0.036), anoxic conditions (*p* = 0.024), or oxic and anoxic conditions (*p* = 0.00016), with sediment communities being more diverse than leachate communities.

### 3.5. Bacterial Community Composition

The cluster dendrogram based on the bacterial 16S rRNA gene diversity shows a clear separation between sediment and leachate samples (Figure 7). It also indicates that leachate samples were more similar amongst themselves than the sediment samples. The first timepoints of the sediment oxic and anoxic incubations cluster together, suggesting that oxygen presence/absence did not play a major role in the community composition in the first 12 h of incubation. All other timepoints, whether in the oxic or anoxic conditions do not cluster together, indicating major community differences. The community composition of the sample collected in the dam lake water (Dam) was most similar to the leachate sample collected after 720 h in anoxic conditions. This suggests that cells leaching from the sediments into the water column explain part of the existing in situ diversity in the Villerest lake water.

Environment type (sediment/leachate) explained 15.4% of the variation in all bacterial samples (PERMANOVA, *p* = 0.001, Table 1), whereas oxygen presence/absence explained 13.8% of the variation (*p* = 0.002), and incubation time explained an additional 15.6% of the variation (*p* = 0.04), confirming observations made with the cluster dendrograms.

The lefse analysis comparing bacterial community composition between oxic and anoxic samples in the sediment, *Sphingomonas*, *Pseudoarthrobacter*, and *Dyella* were significantly higher in the oxic community, whereas *Bacillus* and *Clostridium* were significantly higher in the anoxic community (Figure 8a). In the leachate, *Methylocella* and *Micrococcus* were significantly higher in the oxic community, whereas *Flavobacterium*, *Cellulomonas*, and *Caulobacter* were significantly higher in the anoxic community. When comparing the sediment and leachate samples in the oxic incubations, *Bacillus* and *Novosphingobium* were significantly higher in the sediments (Figure 8b). In the anoxic incubations, *Bacillus* and *Clostridium* were significantly higher in the sediments, and *Sphingomonas*, *Flavobacterium*, and *Magnetospirillum* were significantly higher in the leachate.

### 3.6. Source of Leachate and Sediment Archaeal and Bacterial Communities

For the Archaea, we only used the sediment dataset for the FEAST analysis, as the number of sequences in the leachate samples were too low. Also, the analysis could only be done for the 240 h and 720 h samples, as the FEAST analysis requires a minimum of two source samples. The portion of unknown sources for the oxic sediment samples was lower (55.8 and 56.2%) than for the anoxic samples (81 and 70%; Figure 9a). For the oxic samples, the initial sediment accounted for 3.6% (after 240 h) and 1.5% (after 720 h) of the communities, whereas this source was negligeable for the anoxic samples (less than 10^−4^). Furthermore, for the oxic incubations, the source of the communities was the community from the previous time point (40.1% after 240 h and 36.8% after 720 h). For the anoxic ones, the source was mainly the community from the previous time point after 240 h (S12H_ax contributed 18.2% to S240H_ax), and after 720 h the source originated from both previous time points’ communities (S12H_ax contributed 6.6% and S240H_ax 23% to S720H_ax).

For the bacterial samples, we were able to run potential sources for the 12 h samples, since we used the leachate samples as a source for the sediment samples and vice-versa. In oxic conditions, a vast majority of the sediment community source was unknown after 12 h incubation (99.9%; Figure 9b). Source for the 240 h samples were the initial sediment, and both leachate samples from the previous and same timepoints. We observed the same for the 720 h, with the major source being the sediment sample from the previous timepoint (17.6%). In anoxic conditions, a small source of sediment community for the 12 h incubation was the leachate sample from the same timepoint (3.1%). After 240 h incubation, a small source was the sediment sample from the previous timepoint (5.1%). And after 720 h, a small source was the water sample from the previous timepoint (W240H_ax contributed 5%), but the major source was the sediment samples from the previous timepoints (S12H_ax contributed 32.6% and S240H_ax 21.7%). For the leachate samples after 12 h whether in oxic or anoxic conditions, most of the sources were unknown (99.9 and 99.4%). For the oxic incubations, after 240 h and 720 h, the 240 h sediment community contributed 8.8% and 4.3% to the community, whereas leachate samples from the previous timepoints were the major sources (W12H_ox contributed 23.1% to W240H_ox, and W240H_ox contributed 24.3% to W720H_ox). For the anoxic incubations, most of the sources remained unknown, with a small source from the previous leachate timepoints (W12H_ax contributed 13.7% to W240H_ax, and W240H_ax contributed 1.3% to W720H_ax).

## 4. Discussion

### 4.1. Organic Matter Degradation and Mineral Alteration

During the leaching experiments, both mineral dissolution/precipitation and organic matter degradation occurred. Oxic conditions favoured the dissolution of minerals such as sulphides and the degradation of organic matter. In contrast, anoxic conditions mainly favoured the dissolution of oxyhydroxide phases and the degradation of organic matter.

In order to evaluate POM degradation, several proxies such as DOC, TDN, P-PO_4_, and potentially S-SO_4_ and protons can be used. DOC concentrations should increase as POM is degraded. However, part of this dissolved organic carbon can be oxidized to CO_2_, leading to a decrease of DOC, and thus, DOC cannot be used as a conservative parameter. Under oxic conditions, TDN can be considered as a good tracer of POM degradation as nitrate cannot be reduced to N_2_ in the presence of dissolved oxygen. Under anoxic conditions, nitrogen originating from POM degradation can exist in its N(-III) form, the only species accounting for DON, and it can be partly oxidized to nitrate by MnO_2_ [46,47] or by an anammox reaction mediated by bacteria [48,49]. In both cases, nitrogen can then be reduced to N_2_. Hence, the TDN concentration cannot be considered conservative under anoxic conditions. Therefore, variations of all nutrients have to be taken into account.

In our study under oxic conditions, pathways of organic matter degradation led to its partial mineralization in CO_2_ as DOC decreased and nitrate formation was still greater than consumption, but a significant consumption of phosphorous limited its input.

Under anoxic conditions different and less mineralized by-products of organic matter degradation were observed. TDN reached a plateau after 10 days. Nitrate consumption may have caused a slowdown in organic matter degradation. Under these conditions, phosphorous was not limiting. DOC production was much greater, whereas less TDN was produced. Anoxic leachates emitted a putrid odour, possibly caused by organosulphur molecules. No phosphorus consumption was clearly evidenced under anoxic conditions (i.e., consumption was lower than production). Hence, under both conditions, sulphate release was recorded, but it was greater under oxic conditions. Under the latter condition, sulphate may have originated from sulphide oxidation as framboidal pyrites were observed in sediment, and this explains why proton input was much greater under oxic conditions.

### 4.2. Microbial Community Changes in the Sediment

During both the oxic and anoxic incubations, bacterial diversity increased with time (from 12 to 240 and 720 h), whereas archaeal diversity was relatively stable, suggesting a higher capacity of the Archaea to acclimatize and respond to the resuspension event mimicked during our incubations. This could also indicate that Bacteria were more vulnerable to the change in their abiotic environment compared to the Archaea, leading to the settling of new additional taxa. These observations could be linked to potential exchanges of nutrients and microbial cells from water to sediment. Indeed, bacterial water communities were a non-negligeable source for the bacterial sediment communities over the three timepoints, likely contributing to the community change and diversity increase. On the other hand, since very few Archaea seem to inhabit the dam lake water (as attested by both the leaching experiments, and the sequencing of the lake water itself), the only source to the archaeal sediment communities were the communities actually or previously present. Therefore, less community and diversity change occurred during the resuspension event.

Dynamic hydraulic events caused, for instance, by dam maintenance, can lead to the flow of oxygenated water from the lake epilimnion, but also anoxic waters when organic matter concentrations are high [24]. For this reason, we investigated the consequences of resuspension in oxic or anoxic conditions. When looking at sediment and water samples separately, the presence/absence of oxygen did not significantly influence archaeal and bacterial diversity indices. However, the presence/absence of oxygen did account for some of the variation observed in the bacterial mesocosm communities. Clustering of bacterial samples showed that after 12 h, both oxic and anoxic samples clustered together, whereas all samples after 12 h did not cluster at all, indicating major community shifts occurring after 12 h when faced with a resuspension event. This suggests that if the hydraulic events have short durations, the water characteristic in terms of oxygen content will not affect the bacterial community. In oxic conditions, the initial sediment used to set up the incubation (S0) only explained a small proportion of the communities for both the archaeal and bacterial sediment incubations. The latter was not observed in anoxic conditions, suggesting that during a resuspension event, when faced with oxygenated water, a portion of the bacterial populations present before the event is able to survive and contribute to the community even after 720 h of resuspension activity. It seems likely that mixing with anoxic waters led to a more significant change in environmental conditions, explaining the more drastic change in microbial community composition.

### 4.3. Potential Microbial Metabolisms in the Sediment

Most of the dominant bacterial and archaeal groups detected at each time point were not the dominant groups in the initial sediments, further showing that the changes caused by the resuspension event favored some taxa over others. For the oxic experiments, apart from *Bacillus* at 12 h, the same dominant bacterial genera were detected at each time point (*Pseudoarthrobacter*, *Massilia*, and *Sphingomonas*), with different relative abundances. Members of these genera are all aerobic chemoorganotrophs using saccharides and organic acids as energy and carbon sources, ultimately leading to CO_2_ emission [50,51,52]. The result of these heterotrophic activities can explain the observed decrease of organic matter, as well as the decrease of pH, which occurred throughout the incubations. At the 240 h and 720 h timepoints, we observed the presence of *Massilia*, containing strains able to survive in environments with heavy metal concentrations [51], which might be correlated to the increase of heavy metals detected in the water samples, indicative of carbonate dissolution.

During the anoxic incubations, *Bacillus* was detected at all time points but was dominant after 12 h incubation. All bacterial groups detected after 240 h incubation were mainly fermenters (*Clostridium*, *Desulfitobacterium*, unclassified Prolixibacteraceae) and chemoorganotrophs (*Lutispora*, unclassified Caloramatoraceae) producing ethanol, acetate, lactate, or propionate [53,54,55,56]. In anaerobic conditions, fermentation is much less effective in terms of energy production than respiration, and also produces less CO_2_. Combined with a slower growth rate, this could explain why organic matter degradation reached a plateau at this time point as well. After 240 h incubation, we detected *Desulfitobacterium*, which is able to reduce sulfite, thiosulfate, or nitrate for use as electron acceptors [53]. Sulfate was present in the water samples and reached a peak at 240 h, most likely available to act as an electron acceptor.

Methanogens dominated both the oxic and anoxic incubations, probably explaining why the presence/absence of oxygen did not significantly impact the archaeal diversity indices. Methanogens can use three types of substrates (acetate, methylamines, and bicarbonate) to produce methane, and we detected genera able to use all of them (*Methanosaeta*, *Methanosarcina*, *Methanoregula*, *Methanocella*) [57,58,59,60]. Methanogenesis is the final step in the mineralization of organic matter in soils and sediments. It is highly likely that they were stimulated by the production of acetate, H_2_, and CO_2_ by the chemoorganotrophic bacteria to further degrade organic matter to methane and CO_2_ (except for the autotrophic methanogens, which only produce methane). Thus, methanogenic activities in the sediments could also account for the decrease of organic matter that was observed in the water over time. In return, bacteria that ferment fatty acids benefit from these activities, since the autotrophic methanogens remove H_2_, which inhibits their activities [61]. Methanogens are typically viewed as strict anaerobes because of their oxygen-sensitive enzymes. However, there is evidence that some methanogens contain defense and repair mechanisms that allow them to survive in suboxic environments [62], which could explain why they dominated the aerobic incubations. It is also possible that methanogens locally inhabited sediment pores that were depleted in oxygen. The *Methanoregula* genus was relatively more abundant in the the oxic incubations after 240 h, which can be correlated to the lower pH measured in these mesocosms compared to the anoxic ones (5.5 to 4.4 in the oxic mesocosms, and 6.5 in the anoxic ones). Indeed, this methanogen contains one strain adapted to oligotrophic and acidic (pH 4.5 to 5.5) environments [60].

Bathyarchaeota have been suggested to degrade extracellular recalcitrant molecules such as aromatic compounds, peptidoglycan, lignin, and cellulose [42,63], which are all compounds typically found in lake sediments such as the Villerest dam lake [32,33]. In addition, strains of the bacterial genus *Dyella* dominant after 240 h are also able to hydrolyze cellulose [64]. The presence of microorganisms able to hydrolyze complex polymers into smaller, more usable molecules most likely stimulated the activities of chemoorganotrophic bacteria during the incubations and participated in the overall decrease of organic matter to CO_2_.

### 4.4. Leaching of Microbial Communities in the Water

The initial water used to set up the incubations was sterile and contained no organisms, hence, the populations that thrived in the leachate samples necessarily stemmed from the sediments collected in the Villerest dam, using nutrients leaching from the sediments. Bacterial diversity increased with time, indicating that some of the benthic bacteria leached into the water and were able to grow and acclimate to a planktonic lifestyle. However, the diversity indices were significantly lower in the leachate samples than in the sediments, suggesting that most of the benthic bacteria could not acclimatize to an aquatic niche. It is also likely that many organisms were missing nutrients, carbon, and energy sources to survive. This is further supported by the fact that the environment type (water/sediment) explained part of the bacterial mesocosm community variance, and that the lefse analysis indicated that a big number of genera were significantly higher in the sediment samples compared to the leachate samples. Variation was smaller inside the water sample group compared to the sediment sample group, but this could also be due to the fact that we did not add nutrients to the water during the leaching experiments.

Bacterial source tracking showed that the water communities stemmed from water communities from previous timepoints. Sediment communities contributed to the water communities mainly in the oxic incubations, suggesting that leaching of bacterial populations from the sediment to the water column would be mostly favorable when resuspension events were caused by oxygenated waters.

The extremely low number of archaeal reads recovered in the leachate samples could be explained by the high number of methanogenic sequences found in the sediments. The methanogens might have endured some protected exposure to oxygen in the sediments, but as soon as they leached in the oxic water it is most likely most did not survive. These results also suggest that benthic archaea do not settle well in an aquatic environment, which is supported by the low number of reads recovered from the dam lake water.

### 4.5. Potential Microbial Metabolisms in the Water

In oxic conditions, most dominant water bacterial species were also dominating the sediments (*Sphingomonas*, *Pseudoarthrobacter*, and *Dyella*). As observed in the sediments, these bacterial groups are chemoorganotrophs assimilating saccharides, organic acids, or long chain alkanes, and producing CO_2_. These chemoorganotrophs probably benefited from the leaching of organic matter from the sediments (due to abiotic or biotic reactions) to the water, and also contributed to decreasing the pH. Moreover, *Sphingomonas* and *Dyella* are acido-tolerant [65,66], allowing them to grow in the waters where the pH decreased overtime (5.5 to 4.4 in the oxic mesocosms).

The presence of the methane oxidizing *Methylocella* after 12 h incubation, and not after, likely indicates that methanogenic activities were highest in the sediments within the first 12 h [67]. Also, this bacterium can fix N_2_, which could partially account for the increase of total dissolved nitrogen (TDN) in the oxic water samples. After 720 h incubation in oxic conditions, there was a high relative abundance of *Rhodanobacter*, which contains strains able to break down lindane [68]. This aromatic compound is a component of insecticides and suggests that appearance of a bacterial population able to use more complex carbohydrates when all simpler ones are depleted. Bacterial degradation of lindane leads to organic acids, which can be used by the other chemoorganotrophic bacteria.

*Sphingomonas*, *Pseudoarthrobacter*, and unclassified Burkholderiaceae were identified after 12 and 240 h incubation in anoxic conditions. These bacteria are chemoorganotrophs that use nitrate as electron acceptor when oxygen is not available. A metal-reducing microbe (*Magnetospirillum*) was identified at the 240 h timepoint [69]. This dissimilatory metal-reducer completely oxidizes short-chain carboxylic acids probably produced by the other chemoorganotrophic bacterial communities. Only chemoorganotrophs composed the dominant bacterial groups after 720 h incubation. Two of the distinct genera detected show evidence of adaptation to oligotrophic growth, suggesting that most nutrients were depleted. Indeed, *Cellulomonas* can use complex recalcitrant polymers derived from plant and/or fungal walls [70], while *Caulobacter* has a unique reproductive mode that reflects adaptation to nutrient-limited environments [71].

Most of the few archaeal reads recovered in the oxic waters belonged to the Bathy_6. Although no isolated cultures of Bathyarchaeota exist, it is thought they are mostly benthic organisms, which would also explain why leaching of members of this group did not survive well in the aquatic environment. Archaeal ammonia oxidizers were detected in the waters and could have supplied the Bacteria and some archaea with nitrate. Indeed, we detected sequences affiliated with the methane oxidizing denitrifying cand. Methanoperendens [72]. This archaeon could have benefited from the activity of the ammonia oxidizing Nitrososphaeraceae producing nitrate [73]. It could also have contributed to the observed dissolved organic nitrogen (DON) increase by producing N_2_.

## 5. Conclusions

Our controlled leaching experiments highlighted mineral dissolution/precipitation and organic matter degradation in both oxic and anoxic conditions. Most of the microorganisms identified during the incubations were either chemoorganotrophs or fermenters, likely acidifying their environment, and participating in carbonate dissolution and trace element release. This has been previously documented using dredged sediments [28]. Resuspension events seemed to favor the activities of microbes involved in organic matter degradation, with a chain of degradation using complex polymers down to C1-type compounds, suggesting a vast microbial network cooperation.

The leaching experiment influenced microbial diversity and community composition in the sediment samples. Archaeal was stable over time, compared to bacterial diversity, indicating a higher vulnerability of Bacteria to fluctuating conditions. We did not detect many Archaea in the leachate water, and bacterial diversity was lower in the leachate samples, suggesting that many of the benthic microbial populations cannot settle or colonize the water column following a resuspension event.

The type of water (oxygenated or anoxic) that causes hydraulic events played a major role in selecting for distinct microbial communities, which, in turn, influenced organic matter degradation and the release of trace elements, probably due to a slower metabolism of anaerobic microbes. It is likely, however, that experimental conditions biased the observed results [74], and future studies based on metagenomic or metatranscriptomic analyses of in situ samples before and after resuspension events will help support our findings.

## Figures and Tables

**Figure 1 genes-13-01416-f001:**
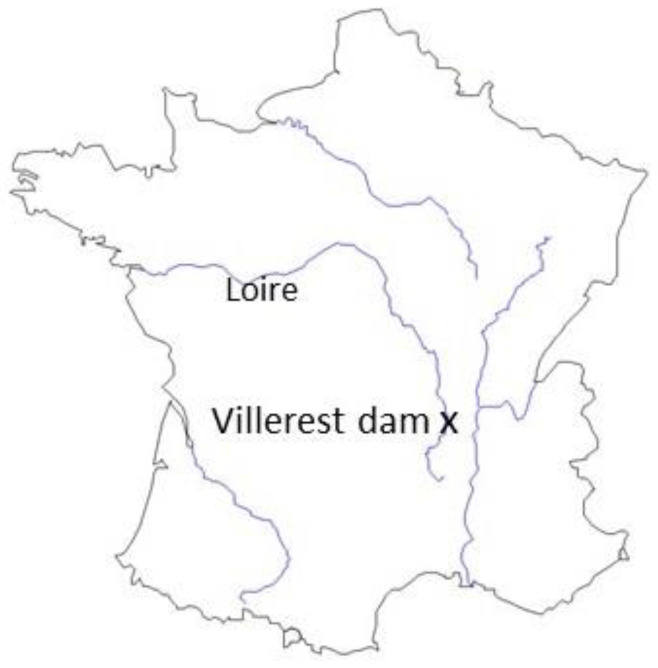
Map of France, locating the Loire River and the Villerest dam.

**Figure 2 genes-13-01416-f002:**
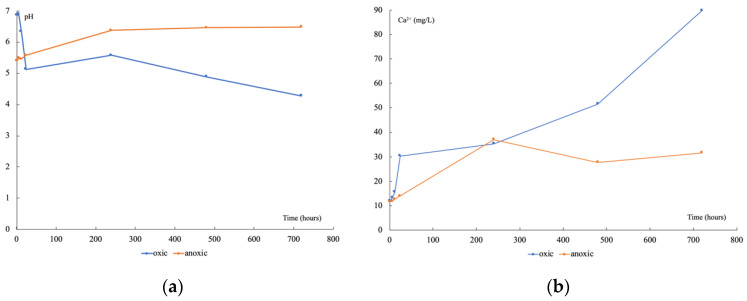
Time variation of pH (**a**), and Ca^2+^ (**b**), during the leachate incubations, in oxic (blue) and anoxic (orange) conditions.

**Figure 3 genes-13-01416-f003:**
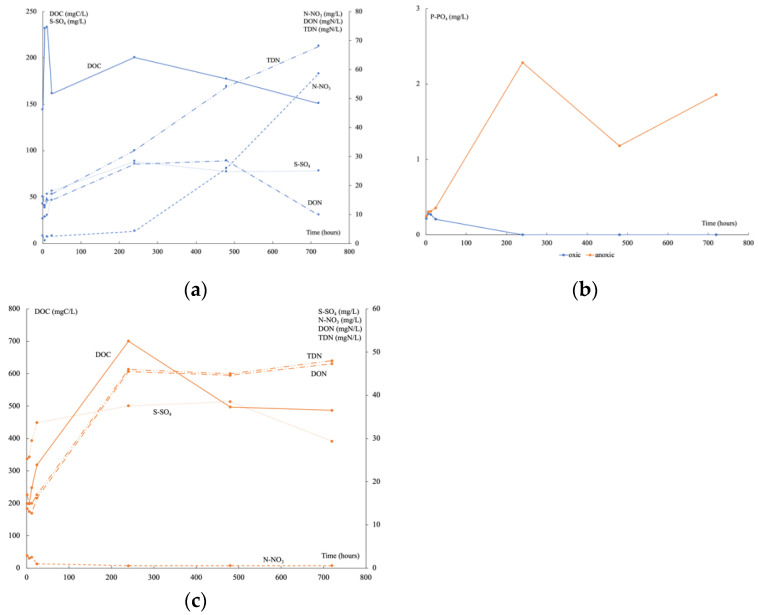
Time variation of DOC, S-SO_4_, N-NO_3_, DON, and TDN in oxic (**a**) and anoxic conditions (**c**), and of P-PO_4_ (**b**) in oxic (blue) and anoxic (orange) conditions.

**Figure 4 genes-13-01416-f004:**
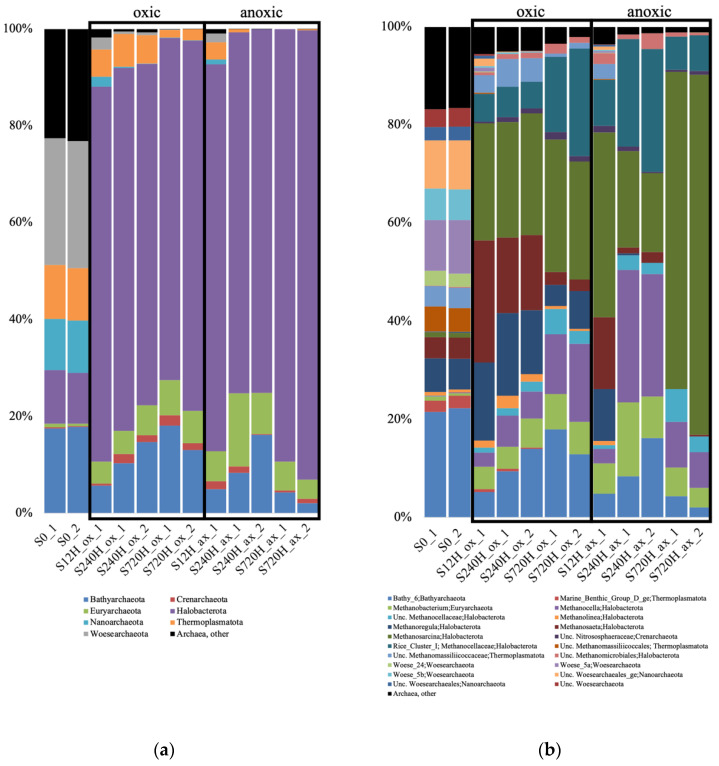
Relative abundance of the archaeal taxonomic groups at the phylum (**a**); and genus (**b**) levels, in the oxic (ox) and anoxic (ax) incubations, in the sediment (S) samples, and for both duplicates (_1 and _2). Relative abundance is expressed in percentage of the total number of sequences for each sample. S0, initial sediment before incubation; Unc, unclassified.

**Figure 5 genes-13-01416-f005:**
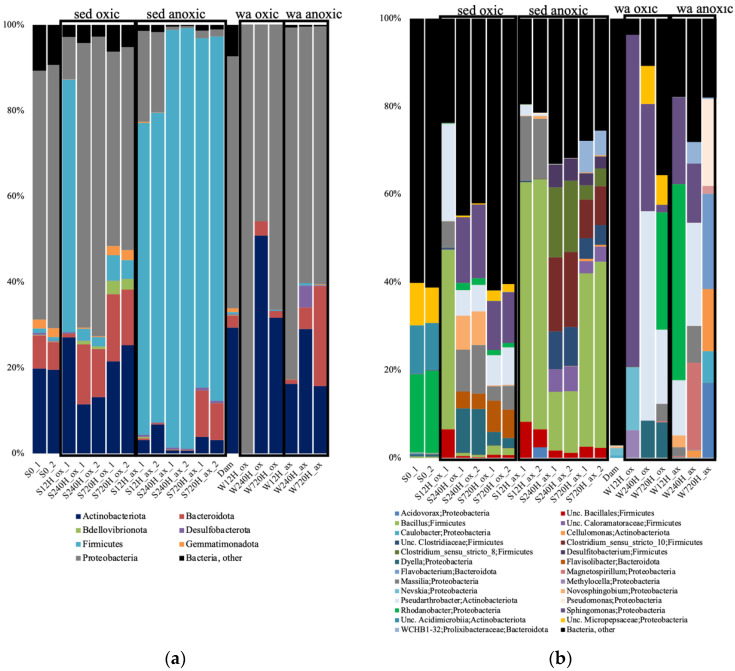
Relative abundance of the bacterial taxonomic groups at the phylum (**a**); and genus (**b**) levels, in the oxic (ox) and anoxic (ax) incubations, in the sediment (S) and water (W) samples and for both duplicates (_1 and _2). Relative abundance is expressed in percentage of the total number of sequences for each sample. S0, initial sediment before incubation; Dam, dam lake water sample; Unc, unclassified; sed, sediment; wa, water.

**Figure 6 genes-13-01416-f006:**
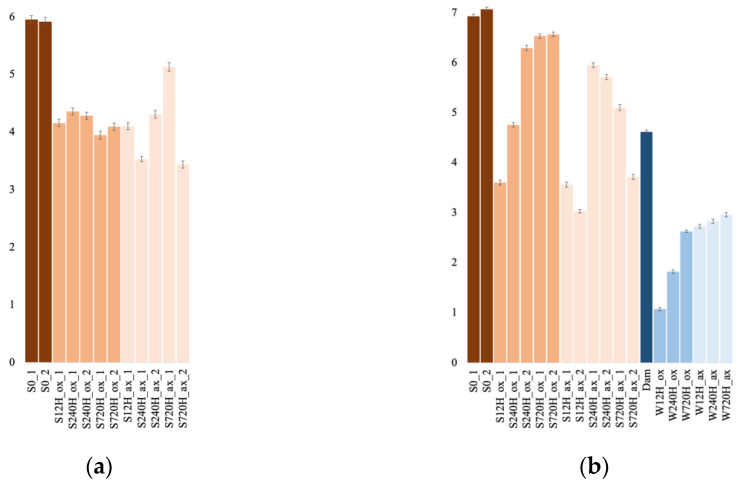
Shannon diversity indices for the archaeal (**a**) and bacterial (**b**) communities, in the oxic (ox) and anoxic (ax) incubations, in the sediment (S) and water (W) samples and for both duplicates (_1 and _2). S0, initial sediment before incubation; Dam, dam lake water sample.

**Figure 7 genes-13-01416-f007:**
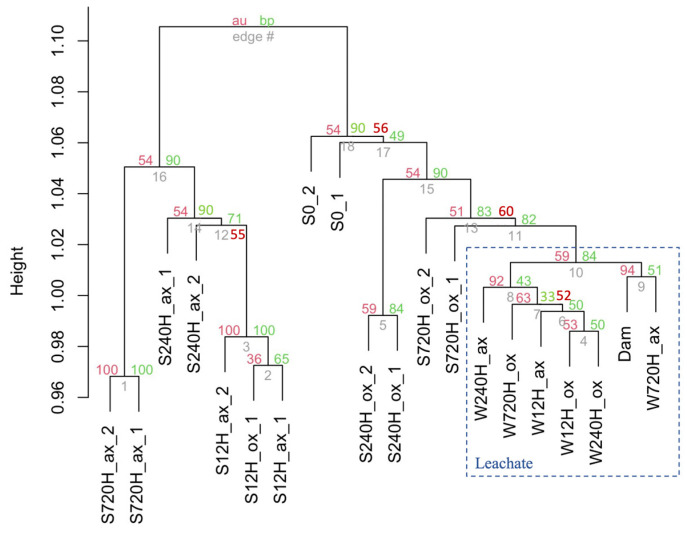
Dendrogram from cluster analysis of bacterial 16S rRNA gene diversity in the oxic (ox), and anoxic (ax) incubations, in the sediment (S) and water (W) samples and for both duplicates (_1 and _2). Bar indicates dissimilarity values. Approximately unbiased (au) p values are shown in red, and bootstrap probability (bp) values are shown in green at each node. S0, initial sediment before incubation; Dam, dam lake water sample.

**Figure 8 genes-13-01416-f008:**
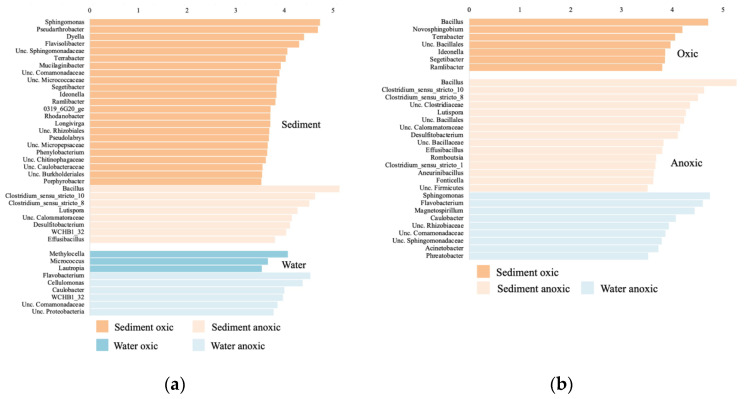
Linear discriminant analysis (LDA) score comparing significantly different bacterial genera between (**a**) the oxic and anoxic samples in the sediment and leachate samples separately; (**b**) the sediment and leachate samples in the oxic and anoxic samples separately; calculated using the lefse analysis. Genera with a LDA score > 3.5 are displayed.

**Figure 9 genes-13-01416-f009:**
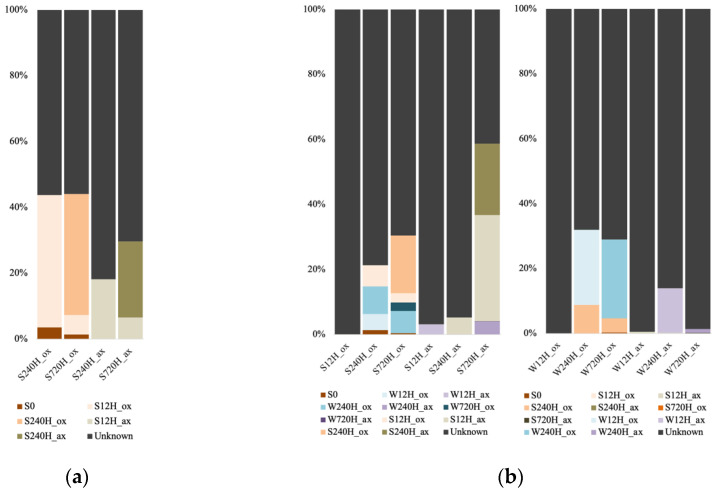
Microbial source tracking of (**a**) archaeal sediment communities using the initial sediment (S0) and the previous sediment incubations time points (S) as sources; (**b**) bacterial sediment and leachate communities, using the initial sediment (S0) and the previous sediment (S) and water (W) incubation timepoints as sources.

**Table 1 genes-13-01416-t001:** Variation in bacterial community composition explained by oxygen (oxic/anoxic incubation conditions), environment type (sediment/leachate), or incubation time (12, 240 or 720 h), tested using PERMANOVA.

	**Df**	**SumOfSqs**	**R2**	**F**	**Pr (>F)**
**Oxygen**	1	0.9022	0.1377	2.9917	0.002
**Environment**	1	1.0081	0.15387	3.3429	0.001
**Incubation time**	2	1.0226	0.15609	1.6956	0.04
**Residual**	12	3.6187	0.55234		
**Total**	16	6.5515	1		

## Data Availability

The obtained sequences were deposited in the National Center for Biotechnology Information (NCBI) under the BioProject ID: PRJNA805122.

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
