# Peer review of "Laboratory-Controlled Experiments Reveal Microbial Community Shifts during Sediment Resuspension Events"

_genes, 2022, doi:10.3390/genes13081416_

Round 1
Reviewer 1 Report
Introduction
Please, add the hypotheses.
Materials and methods
The first sentence (L. 66067) is not necessary, since below you describe both study site and sampling methods.
L. 79-81 – So, you took water samples to compare them with sediment leachates? If yes, you should add such information.
L. 82- How many replications you had? Seven cores were taken, so there were seven replications for each condition?
L. 96-97 – It looks like these parameters are not important since you do not present them in this ms, So, this sentence is unnecessary.
L. 98-99 – So, these figures will be presented in the paper by Shumkysh et al. which is currently prepared? This information is weird. All the data necessary to explain and discuss your finding should be presented in you ms.
L. 101 – When samples for DNA analyses were taken? Prior incubation, and then after 24, 240 and 720 hours?
Results
L. 220 – Is ‘lefse’ a formal name of this analysis? Probably is the name of the software, whereas the analysis is named ‘linear discriminant analysis’.
L. 235-273 – This paragraph might be quite surprising for the potential reader. You do not mention it in Material and methods. Please, describe hoy you estimated the sources for microbial communities and what was the purpose of these estimations.
Line 242-243 – 1.10-4 is a negative number. Maybe you thought about 1·10-4. Moreover, unit (%) is missing.
Discussion
Line 227-228 – ‘…a higher capacity of the bacteria to acclimatize…’ versus L. 279-280 ‘…Bacteria were more vulnerable to the changes in their abiotic environment…’. I feel some conflict between these two sentences. It looks like Archaea had higher capacity to acclimatize, whereas some groups of bacteria might decrease and their niches were settled by other groups (=increase in diversity).
L. 400 – ‘…many, if not all these groups…’ – too general, informal and presumed. Please, rephrase.
Conclusions are not obligatory in this journal. However, as your ms presents several novel information it would be advisable to get them together in the conclusions. You also should try to explain whether and how the environment of your experiment might alter the responses of microbial communities to the studied events. You can see the paper by Adamczuk (Do experimental conditions bias plankton responses to increased concentration of dissolved organic matter (DOM)? A meta-analytical synthesis of the published results) showing how experimental conditions alter the responses of plankton to organic matter.
Author Response
Introduction: Please, add the hypotheses.
Answer from the authors: 3 hypotheses were added at the end of the introduction, L74-77.
Materials and methods
The first sentence (L. 66-67) is not necessary, since below you describe both study site and sampling methods.
Answer from the authors: the sentence (and information) was deleted.
- 79-81 – So, you took water samples to compare them with sediment leachates? If yes, you should add such information.
Answer from the authors: actually, the in situ water was sampled from the lake, in order to compare with the water community that was observed in the incubations after leaching of cells into the artificial water column. To make this clearer in the paper, we added a part to the sentence L93-95.
- 82- How many replications you had? Seven cores were taken, so there were seven replications for each condition?
Answer from the authors: the seven cores were collected in order to gather enough material to set up the incubations, and to have a homogenized sample since lake sediments tend to be heterogeneous. So the sediments were mixed and the seven cores are not replicate samples.
- 96-97 – It looks like these parameters are not important since you do not present them in this ms, So, this sentence is unnecessary.
Answer from the authors: we revised the paper in order to now incorporate the data in this paper. Please see L139-156, L201-228 and L366-396.
- 98-99 – So, these figures will be presented in the paper by Shumkysh et al. which is currently prepared? This information is weird. All the data necessary to explain and discuss your finding should be presented in you ms.
Answer from the authors: Please refer to our previous answer.
- 101 – When samples for DNA analyses were taken? Prior incubation, and then after 24, 240 and 720 hours?
Answer from the authors: yes indeed, sediment samples were taken prior to the incubations, and then after 12, 240, and 720 hours. This information was added to the text L160-161.
Results
- 220 – Is ‘lefse’ a formal name of this analysis? Probably is the name of the software, whereas the analysis is named ‘linear discriminant analysis’.
Answer from the authors: as indicated in the material and methods (L194), lefse stands for linear discriminant analysis effect size, which is the full name of the analysis used. Please refer to the introduction on the Huttenhower lab website (https://huttenhower.sph.harvard.edu/lefse/).
- 235-273 – This paragraph might be quite surprising for the potential reader. You do not mention it in Material and methods. Please, describe hoy you estimated the sources for microbial communities and what was the purpose of these estimations.
Answer from the authors: actually, the method used is described in the material and methods L195-199. However, we agree the sentence might not be clear enough, so we added a few words of explanation L198.
Line 242-243 – 1.10-4 is a negative number. Maybe you thought about 1·10-4. Moreover, unit (%) is missing.
Answer from the authors: indeed, thank you for catching this. We corrected this L335.
Discussion
Line 227-228 – ‘…a higher capacity of the bacteria to acclimatize…’ versus L. 279-280 ‘…Bacteria were more vulnerable to the changes in their abiotic environment…’. I feel some conflict between these two sentences. It looks like Archaea had higher capacity to acclimatize, whereas some groups of bacteria might decrease and their niches were settled by other groups (=increase in diversity).
Answer from the authors: we thank the reviewer for this comment. This is indeed a sound suggestion and changes were made L400-403.
- 400 – ‘…many, if not all these groups…’ – too general, informal and presumed. Please, rephrase.
Answer from the authors: the change was made L531.
Conclusions are not obligatory in this journal. However, as your ms presents several novel information it would be advisable to get them together in the conclusions.
Answer from the authors: we thank the reviewer for her/his suggestion and modified the last paragraph to make it about concluding remarks L571-591.
You also should try to explain whether and how the environment of your experiment might alter the responses of microbial communities to the studied events. You can see the paper by Adamczuk (Do experimental conditions bias plankton responses to increased concentration of dissolved organic matter (DOM)? A meta-analytical synthesis of the published results) showing how experimental conditions alter the responses of plankton to organic matter.
Answer from the authors: we thank the reviewer for her/his suggestion, and added a sentence at the end of the conclusions to highlight this possibility L589-591.
Reviewer 2 Report
In the manuscript entitled "Laboratory-controlled experiments reveal microbial community shifts during sediment resuspension events", the authors describe the changes in microbial community in a temporal study of sediment resuspension under different oxygen conditions. The manuscript is well written and describes a scientifically-sound study with a high degree of novelty. However, in my opinion part of the message is missing, as the authors chose to exclude the analysis of the soil chemistry and abiotic variables (how these affect the variations in community structure and composition) for a follow-up paper. I would suggest that by including these analyses in the manuscript, the authors would make a much stronger argument for the discussion of the results, and would increase the overall impact of the study. In addition, please see below some queries/corrections regarding the paper:
Introduction
Lines 35-37: References missing for these statements.
Lines 42-45: Missing references for this statement.
Lines 46-47: Do not understand what the authors mean by “leading to concentration levels that may cause sanitary problems for riparian populations”. Do the authors mean that the concentration of trace elements might become harmful to the local biodiversity? Please specify. In addition, please include the suitable references for this statement.
Line 50: Please do not include references that are in preparation if the readers have no access to the information.
Lines 61-64: In my opinion, these statements are misplaced. Perhaps the authors might want to include a small paragraph on the mixing process of oxic and anoxic water in the lake ecosystem earlier in the introduction. Otherwise, I would suggest deleting these sentences and including a final statement to highlight the novelty and ecological importance of the study.
Materials and Methods
Lines 66-81: It is unclear why the authors opted to sample the sediment in 2016 and the water in 2017 instead of at the same time. Wouldn’t the time gap affect the analysis of the sediment vs the water biodiversity, since the communities might have changed during that period? Please include a sentence or two to explain the rationale of this sampling strategy.
Lines 82-99: Please clarify in the text what sediments and water samples were used for the batch experiments.
Lines 98-99- Please include the number of the figures (eg. Figure S1, etc).
Lines 110-111: Please include the sequences of the primers with the suitable references.
Lines 111-112: Please include more details regarding the quality trimming and processing of the reads. What quality score was chosen? How many bps were truncated? Etc…..
Lines 118-135: Missing reference for the statistical analyses performed (eg. PERMANOVA) and the r packages used.
Discussion
Lines279-281: Since the authors do not include any nutrient/soil chemistry parameters in the analysis, I would be very cautious about making these statement. If the authors have nutrient and soil chemistry data on the sediments and leachates across the different incubation times, I would suggest to include a result section investigating the main abiotic drivers of community shift other than time and oxygen levels. Considering one of the main objectives of the study is to “assess the role of the in situ microbial community in MTE release”, I would expect the authors to include data on MTE in the leachate or sediments. In fact, the authors seem to indicate later in the manuscript (lines 316-319) that they have the soil chemistry data for the experiment, and therefore I would suggest for them to include it as part of the analysis of the abiotic factors that might affect microbial composition and vice-versa (correlation between community shifts and abiotic shifts).
Lines 281-283: See my comment above regarding the collection of sediment and water samples at different time points. Do the authors expect the samples to be comparable, considering that the ecosystem might have changed dramatically in a space of a year?
Lines 289-291: These statement could be re-arranged to the introduction.
Figure 1: For readability, I would suggest for the authors to cluster the samples more clearly into the oxic and anoxic groups, either by highlighting the sample names with different colors depending on the condition or by using something like a face-wrap in Rstudio.
Figure 2: Same suggestion as Figure 1.
Author Response
In the manuscript entitled "Laboratory-controlled experiments reveal microbial community shifts during sediment resuspension events", the authors describe the changes in microbial community in a temporal study of sediment resuspension under different oxygen conditions. The manuscript is well written and describes a scientifically-sound study with a high degree of novelty.
However, in my opinion part of the message is missing, as the authors chose to exclude the analysis of the soil chemistry and abiotic variables (how these affect the variations in community structure and composition) for a follow-up paper. I would suggest that by including these analyses in the manuscript, the authors would make a much stronger argument for the discussion of the results, and would increase the overall impact of the study. In addition, please see below some queries/corrections regarding the paper:
Answer from the authors: we thank the reviewer for her/his encouraging comments and for her/his suggestions as well. Unfortunately, the geochemical parameters were only measured in the leachate water samples. Since we were not able to produce replicates for the water samples, 3 samples are not sufficient to run multivariate analyses. However, we did alter the flow of the manuscript to include the geochemical data that was presented initially as part of a separate paper. We hope this helps with the impact of the study, nonetheless.
Introduction
Lines 35-37: References missing for these statements.
Answer from the authors: references were added L28-45.
Lines 42-45: Missing references for this statement.
Answer from the authors: please see the above answer.
Lines 46-47: Do not understand what the authors mean by “leading to concentration levels that may cause sanitary problems for riparian populations”. Do the authors mean that the concentration of trace elements might become harmful to the local biodiversity? Please specify. In addition, please include the suitable references for this statement.
Answer from the authors: this sentence was deleted from the main text, since we have shifted the focus of the study away from MTEs.
Line 50: Please do not include references that are in preparation if the readers have no access to the information.
Answer from the authors: as stated above, this has been addressed. We have added text L139-156, L201-228 and L366-396.
Lines 61-64: In my opinion, these statements are misplaced. Perhaps the authors might want to include a small paragraph on the mixing process of oxic and anoxic water in the lake ecosystem earlier in the introduction. Otherwise, I would suggest deleting these sentences and including a final statement to highlight the novelty and ecological importance of the study.
Answer from the authors: We added a paragraph with references to better introduce this notion L50-58. Novelty was highlighted in L67-69. Also, in agreement with a comment from reviewer 1, we added hypotheses at the end of the introduction.
Materials and Methods
Lines 66-81: It is unclear why the authors opted to sample the sediment in 2016 and the water in 2017 instead of at the same time. Wouldn’t the time gap affect the analysis of the sediment vs the water biodiversity, since the communities might have changed during that period? Please include a sentence or two to explain the rationale of this sampling strategy.
Answer from the authors: it is true community composition may vary from 1 year to another, however sampling was done both times during the fall season, which is the most stable in terms of community composition in lakes. We added this information in the text L94-96.
Lines 82-99: Please clarify in the text what sediments and water samples were used for the batch experiments.
Answer from the authors: in response to a comment from reviewer 1, this information was added L160-161.
Lines 98-99- Please include the number of the figures (eg. Figure S1, etc).
Answer from the authors: these figures now appear in the main text (Figures 2 and 3).
Lines 110-111: Please include the sequences of the primers with the suitable references.
Answer from the authors: this was added L169-171.
Lines 111-112: Please include more details regarding the quality trimming and processing of the reads. What quality score was chosen? How many bps were truncated? Etc…..
Answer from the authors: this is all indicated on the mothur website, and thus we believe adding this information add unnecessary weight and text to the paper. We added information L172-173 on the read length.
Lines 118-135: Missing reference for the statistical analyses performed (eg. PERMANOVA) and the r packages used.
Answer from the authors: references were added throughout the paragraph.
Discussion
Lines279-281: Since the authors do not include any nutrient/soil chemistry parameters in the analysis, I would be very cautious about making these statement. If the authors have nutrient and soil chemistry data on the sediments and leachates across the different incubation times, I would suggest to include a result section investigating the main abiotic drivers of community shift other than time and oxygen levels. Considering one of the main objectives of the study is to “assess the role of the in situ microbial community in MTE release”, I would expect the authors to include data on MTE in the leachate or sediments. In fact, the authors seem to indicate later in the manuscript (lines 316-319) that they have the soil chemistry data for the experiment, and therefore I would suggest for them to include it as part of the analysis of the abiotic factors that might affect microbial composition and vice-versa (correlation between community shifts and abiotic shifts).
Answer from the authors: as indicated above, we have shifted the focus of the study based on the data we have. We will present the geochemical parameters, which will be discussed L366-396. However, since we cannot run multivariate analyses such as a db-RDA, we will focus on the influence of oxygen as one qualitative factor which can influence community structure. Indeed, there are probably other factors.
Lines 281-283: See my comment above regarding the collection of sediment and water samples at different time points. Do the authors expect the samples to be comparable, considering that the ecosystem might have changed dramatically in a space of a year?
Answer from the authors: please refer to our answer above.
Lines 289-291: These statement could be re-arranged to the introduction.
Answer from the authors: as indicated above, we added some sentences in the introduction.
Figure 1: For readability, I would suggest for the authors to cluster the samples more clearly into the oxic and anoxic groups, either by highlighting the sample names with different colors depending on the condition or by using something like a face-wrap in Rstudio.
Answer from the authors: the figure was modified accordingly.
Figure 2: Same suggestion as Figure 1.
Answer from the authors: the figure was modified accordingly.
Round 2
Reviewer 2 Report
In the revised manuscript, the authors have addressed my major concerns regarding the absence of chemical data and the discussion. The added sections improve the clarity and scientific robustness of the paper. There are only a couple of clarifications I would wish the authors to address:
- In the "Chemical analysis of water leachates" section of the methodology, please specify the time points for the measures, the quantities of input used for the analyses, and whether the measurements were done in triplicates.
- In Figure 2 and 3, please indicate the standard deviations for the measurements at each time point.
Author Response
- In the "Chemical analysis of water leachates" section of the methodology, please specify the time points for the measures, the quantities of input used for the analyses, and whether the measurements were done in triplicates.
- In Figure 2 and 3, please indicate the standard deviations for the measurements at each time point.
Response from the authors: triplicate measurements were not done since sediments were mixed before the incubations in order to have enough volume for the experiment. Details have been added to the text to make the methods clearer L113-131.